# The Role of Resveratrol in Human Male Fertility

**DOI:** 10.3390/molecules26092495

**Published:** 2021-04-24

**Authors:** Laura M. Mongioì, Sarah Perelli, Rosita A. Condorelli, Federica Barbagallo, Andrea Crafa, Rossella Cannarella, Sandro La Vignera, Aldo E. Calogero

**Affiliations:** Department of Clinical and Experimental Medicine, University of Catania, Via S. Sofia 78, 95123 Catania, Italy; lauramongioi@hotmail.it (L.M.M.); sarah.perelli@libero.it (S.P.); rosita.condorelli@unict.it (R.A.C.); federica.barbagallo11@gmail.com (F.B.); crafa.andrea@outlook.it (A.C.); rossella.cannarella@phd.unict.it (R.C.); sandrolavignera@unict.it (S.L.V.)

**Keywords:** resveratrol, human male fertility, sperm parameters, cryopreservation

## Abstract

Resveratrol (RSV) (3,4′,5 trihydroxystilbene) is a natural non-flavonoid polyphenol widely present in the Mediterranean diet. In particular, RSV is found in grapes, peanuts, berries, and red wine. Many beneficial effects of this molecule on human health have been reported. In fact, it improves some clinical aspects of various diseases, such as obesity, tumors, hypertension, Alzheimer’s disease, stroke, cardiovascular diseases, and diabetes mellitus. However, little is known about the relationship between this compound and male fertility and the few available results are often controversial. Therefore, this review evaluated the effects of RSV on human male fertility and the mechanisms through which this polyphenol could act on human spermatozoa.

## 1. Introduction

Resveratrol (RSV) (3,4′,5 trihydroxystilbene) is a natural, non-flavonoid polyphenol widely present in the Mediterranean diet and, particularly, in grapes, peanuts, berries, and red wine [1,2]. However, its dietary intake is low, estimated to be only about 100 µg/day [3]. RSV is well absorbed, rapidly metabolized, and eliminated mainly through the urine [4]. This polyphenol belongs to dietary stilbenes, a class of natural compounds that have significant biological activities of medical interest. It derives from phenylalanine through the activation of the enzyme stilbene synthase and exists in two isomeric forms, *trans*- and *cis*-RSV [5], but the former is the most common.

RSV is a phytoalexin whose biological function is to protect plants from parasitic attack or environmental stress [6]. This molecule also has beneficial effects on human health. A recent review on clinical trials employing RSV has shown that it improves some clinical aspects in patients with obesity, malignancies (colorectal cancer and breast cancer), hypertension, Alzheimer’s disease, stroke, cardiovascular diseases, and diabetes mellitus [7]. Furthermore, clinical data suggest that RSV is safe even at high doses. Indeed, the administration of micronized RSV in patients with cancer did not lead to serious side effects [8]. Furthermore, it is associated with the “French paradox”. In the early 1990s, epidemiological data showed a low prevalence of coronary disease and, generally, a long life expectancy in French people despite a diet characterized by high saturated fat intake. This paradox was attributed to the moderate consumption of red wine with its anti-oxidative effects due to the high redox properties of phenolic hydroxyl groups, which act as free radical scavengers [9].

Although many studies have investigated the beneficial effects of RSV on experimental animals, there are few clinical trials in humans on the relationship between this compound and male fertility and the results are often controversial [10]. A recent review evaluated the impact of RSV on male and female reproduction, by analyzing studies both on human and animal models [10]. The authors concluded that, although a considerable amount of research supports the positive impact of RSV on human and animal reproduction, further studies are necessary to consolidate the knowledge on the properties of RSV and its role in reproductive functions.

Infertility is a widespread condition in industrialized countries where it affects up to 15% of couples of childbearing age [11]. It is defined as the inability to conceive after 1–2 years of unprotected sexual intercourse [12]. Male factor infertility accounts, cumulatively, for half of the couple’s infertility, being solely present in 30% of couples. Idiopathic male infertility is clinically diagnosed after excluding all other known causes of infertility and it affects up to 25% of patients [13]. The management of these patients remains challenging. Most of the studies on infertility treatment, both in vitro and in vivo, have focused on oxidative stress mechanisms which could cause: (1) lipid peroxidation, with alteration of membrane fluidity and permeability resulting in decreased sperm motility and in a reduced capability of spermatozoa to interact with the oocyte; (2) protein modification which causes a reduction of ATP production; and (3) increased sperm DNA fragmentation [14]. Giving its proven anti-oxidative effects, RSV may be useful in the treatment of these conditions.

Therefore, this review evaluates the effect of RSV on human male fertility and the mechanisms by which it could act on human spermatozoa.

## 2. Resveratrol and Human Fertility: Possible Mechanisms of Action

In recent years, many authors have investigated the effects of RSV in male fertility, reporting an improvement in spermatogenesis, including sperm differentiation and number. However, the exact mechanisms by which RSV carries out its beneficial effects on conventional and biofunctional sperm parameters are not fully understood.

Given its structural similarities with diethylstilbestrol (DES) and 17ß-estradiol and its activity as a modulator of the estrogen-response (ER) systems, RSV has been classified as a phytoestrogen [15]. Some studies [16,17,18] have shown that this compound appears to be able to enhance the estrogenic effects of hormones and, therefore, it is a modulator of the female reproductive function. Estrogens are also secreted by the Leydig cells of the human testis where they play a paracrine regulatory function [19,20]. This suggests a possible role for RSV in male fertility. Furthermore, RSV seems to have a phytoestrogen activity in the androgen receptor (AR) by inhibiting its dimerization [21] and its IL-6 induced transcriptional activity [22] in prostate cancer cells. In testis, the activity of RSV may involve some transcription factors that affect the function of the AR [23].

RSV is also the most potent natural compound that activates sirtuin 1 (SIRT-1), the most conserved mammalian NAD^+^-dependent protein and a member of the SIRT family, which may account for its many metabolic benefits in humans [24]. SIRT-1 belongs to human deacetylase and it negatively regulates the p53 tumor suppressor gene promoting cell survival [25].

Another important mechanism is the anti-oxidative activity of RSV, which is due to many different pathways. First, it activates anti-oxidant enzymes, such as catalase and superoxide dismutase [26], which are involved in lipid damage of human spermatozoa during the cryopreservation process [27]. Furthermore, RSV shows an anti-inflammatory activity by decreasing the activity of inflammatory molecules, such as cyclooxygenase 2 (COX2), inducible NOS (iNOS), and nuclear factor kappa-light-chain-enhancer of activated B cells (NF-kB) [28]. It also protects cells from DNA damage and apoptosis by modulating anti- and pro-apoptotic mediators [29,30].

RSV also modulates the expression and activity of multiple drug-metabolizing enzymes. In vitro, it inhibits the enzymatic activity of various cytochrome P450s and blocks their transcription through antagonism of the aryl hydrocarbon receptor (AHR), suggesting that RSV could decrease the exposure of cells to carcinogens [31].

Many evidence suggests that RSV activates adenosine monophosphate-activated protein kinase (AMPK) [24,31,32], which is a serine/threonine kinase that controls cell metabolism by both stimulating catabolic processes and inhibiting anabolic processes [33]. Recently, AMPK has been identified in spermatozoa where it regulates sperm motility, maintains the quality of spermatozoa during long-time storage, and the fertilizing capacity of frozen mouse spermatozoa. This suggests that AMPK can act as a fundamental molecule that allows spermatozoa to adapt to cryopreservation [34]. Studies on AMPK activation by RSV have suggested a variety of apparently contradictory mechanisms. One possibility is an increase in the AMP/ATP ratio. Other studies have suggested that AMPK activation by RSV depends on upstream serine/threonine kinases and calcium/calmodulin-dependent protein kinase β. AMPK can also be activated by reactive oxygen species (ROS), independently of the AMP/ATP ratio, increasing intracellular NAD^+^, a substrate of SIRT-1 [35]. Notably, activation of AMPK has been reported to reduce intracellular ROS levels [36].

## 3. Resveratrol and Human Sperm Parameters

RSV is provided by a peculiarity; at low doses, it improves cell survival, while at high doses, it has cytotoxic effects that are useful in cancer treatment [37]. To assess both these properties, Collodel and collaborators conducted in vitro studies on human spermatozoa [38]. The authors evaluated the motility of swim-up selected spermatozoa to test the effect of different RSV concentrations (from 6 to 100 µM). They found that the maximal effect was reached at the concentrations of 6 and 15 µM of RSV, whereas at the concentration of 100 µM, RSV completely inhibited sperm motility, thus showing cytotoxic effects at higher doses. These data agree with those published subsequently by Nashtaei and co-authors [34]. These authors incubated semen samples with RSV at concentrations of 5, 15, and 25 mM and found that only at the highest concentration tested (25 mM) RSV significantly reduced total and progressive sperm motility (by about 11.5% and 14%, respectively) without any significant effect on the following sperm motility parameters evaluated by computer-assisted sperm analysis: curvilinear velocity, straight-line velocity, average path velocity, linearity of the curvilinear trajectory, and straightness.

Furthermore, Collodel and collaborators [38] investigated also the protective effect of RSV against oxidative stress. This latter is an important factor in the etiology of poor sperm function because it causes peroxidative damage to the cell membrane, DNA fragmentation, and protein oxidation [38,39,40]. The authors tested RSV at the concentrations of 6 and 15 µM to assess its effects against lipid peroxidation (LPO) induced by *tert*-butyl hydroperoxide in human spermatozoa. At the concentration of 6 µM, RSV did not show any protective effect against LPO, whereas at the concentration of 15 µM exhibited a defensive effect. RSV was particularly effective in preserving sperm chromatin texture, but it was not able to fully preserve acrosome, the most fragile sperm organelle. The authors concluded that RSV could have a role as a ROS scavenger in presence of increased oxidative stress, such as during cryopreservation or IVF-ICSI [38].

It is well-known that cigarette smoking and environmental pollution negatively impact male fertility. We have [41] investigated the role of RSV on human sperm damage caused by benzo-α-pyrene (BaP), a polycyclic aromatic hydrocarbon originating from the incomplete combustion of fossil fuels, tobacco smoke, diesel exhaust, and broiled foods [42]. Metabolically activated BaP increases ROS production and consequently the oxidative stress, resulting in increased lipid peroxidation, and caspase, and endonuclease activation [43,44]. We studied the effect of BaP and RSV on human spermatozoa in vitro by assessing total and progressive sperm motility and biofunctional sperm parameters. The latter was evaluated by flow cytometry. We initially tested the effects of BaP alone and found that it significantly inhibited both total and progressive sperm motility in a concentration-dependent manner. Moreover, BaP significantly increased the percentage of spermatozoa with abnormal chromatin compactness, whereas it did not have any significant effect on sperm viability, mitochondrial membrane potential (MMP), and DNA fragmentation. Finally, BaP significantly increased LPO and the production of mitochondrial superoxide. Subsequently, we evaluated the effects of RSV alone and during incubation with BaP on human spermatozoa. RSV alone slightly but significantly decreased the percentage of spermatozoa with abnormal chromatin compactness, did not have any significant effect on LPO, and decreased the percentage of spermatozoa with elevated mitochondrial superoxide production. Moreover, RSV at the concentration of 15 µM/mL significantly counteracted the detrimental effects of BaP on chromatin compactness and LPO. Thus, we concluded that RSV could be considered as a therapeutic option in selected cases of patients with idiopathic infertility. However, further studies were necessary [41].

Recently, Illiano and colleagues published a prospective single-center clinical study to evaluate the effects of a nutraceutical based on RSV (300 mg) but also containing vitamin D, B6, B12, and folic acid on sperm parameters of patients with idiopathic infertility [14]. The authors enrolled 20 patients with oligozoospermia and/or with asthenozoospermia with six months of follow-up after treatment. They found that RSV plus multivitamin supplementation significantly improved sperm concentration, total sperm count, and total and progressive motility. Sperm morphology, pH, and seminal fluid volume did not change after treatment [14]. The authors concluded that the possibility of targeting the metabolic and energetic pathways involved in spermatogenesis and mitochondrial activity with a molecule such as RSV, could lead to potential effects and counteract subfertility/infertility in men through a mitochondrial dynamic mechanism.

Table 1 summarizes the main studies exploring the effects of RSV on human conventional and biofunctional sperm parameters.

## 4. Effects of Resveratrol on Sperm Cryopreservation

Cryopreservation is a routinely used procedure to store spermatozoa to be used in cycles of assisted reproductive techniques (ART) [52]. However, it damages a proportion of spermatozoa that do not survive thawing or have an alteration of their functional capacity. This is probably due to the excessive production of ROS and/or a reduced antioxidant capacity of the freezing medium, leading to increased oxidative stress [53]. For this reason, many studies have investigated whether RSV, thanks to its antioxidant effects, could counteract the effect of oxidative stress if added to semen samples before freezing.

In 2009, Branco and co-authors [45] evaluated the efficacy of RSV or ascorbic acid in preventing DNA damage induced by cryopreservation in spermatozoa of 10 infertile patients compared to 10 fertile controls. Specifically, before cryopreservation, each semen sample was divided into three aliquots: control, 10 mM of RSV, and 10 mM of acid ascorbic. The authors found that cryopreservation significantly increased DNA damage in all groups, although it was bigger in infertile patients, and ascorbic acid addition could prevent it only in the infertile group. On the contrary, RSV at a dose of 10 mM significantly decreased DNA damage in both groups of men studied, but it was not able to prevent the post-freezing motility decrease, so it was hypothesized that other factors apart from oxidative stress influenced post-cryopreservation motility reduction. Lower concentrations (0.1 and 1 mM) of RSV did not have significant effects on sperm DNA damage after freezing. The same authors got similar results in a subsequent study published in 2010 [46]. This study showed that RSV at different concentrations (0.1, 1, and 10 mM) was efficacious in preventing sperm lipid damage induced by cryopreservation in both fertile men and infertile patients in a dose-independent manner. On the contrary, the authors confirmed that RSV did not protect spermatozoa from losing motility [46].

Similar results were published by Meamar and collaborators [47]. These authors showed that the addition of RSV to the cryopreservation medium had different effects according to the concentration used. As suggested by previous studies, spermatozoa with DNA fragmentation were divided into two groups, based on the avidity for the nuclear probe propidium iodide (PI): PI ^brighter^ (PI^br^), consisting of spermatozoa with partially fragmented DNA, and PI^dimmer^ (PI^dim^), consisting of dead spermatozoa with entirely fragmented DNA. The authors after cryopreservation showed a statistically significant increase of sperm DNA fragmentation (SDF, evaluated by TUNEL assay) only in the PI^br^ population. In this population, RSV induced a slight, but significant decrease of post-thawing sperm DNA fragmentation levels compared to controls, at concentrations of 10 and 100 µM, without significantly affecting sperm motility or viability. Conversely, motility and viability were statistically significantly reduced when RSV was used at a concentration of 1 mM. These findings suggest that RSV at high concentrations also has toxic effects during cryopreservation, as shown on fresh semen samples [38]. The peculiarity of this study was that the protective effect of RSV was obtained at a much lower concentration than that used in previous studies (10 µM vs. 10 mM). However, the authors underlined that the effect of RSV (and other antioxidant substances used in their study) was very slight and it was not sufficient to completely prevent the damage induced by the cryopreservation process, suggesting other possible mechanisms involved in sperm damage, as an increase of apoptotic processes [47].

Li and collaborators published a study with opposite results [48]. They found that in comparison with the non-treated group, the post-thaw sperm cryopreserved with 30 μmol/L of RSV showed markedly higher progressively motile sperm, total motility, and viability.

RSV may improve cryopreserved sperm functions by activating 5′-AMPK. Shabani Nashtaei and co-authors [34] reported that AMPK is expressed in human fresh spermatozoa and it is mainly localized in the post-equatorial region of the sperm head and all along the entire flagellum. They incubated spermatozoa from normozoospermic donors for 30 or 60 min with different concentrations of Compound C (1, 10, 30 µM), an AMPK inhibitor, or RSV (5, 15, 25 µM), that acts as AMPK activator, as previously discussed. Subsequently, the semen samples were cryopreserved. The authors showed that RSV significantly increased AMPK phosphorylation and MMP, whereas it decreased ROS and apoptosis-like changes in frozen-thawed spermatozoa. Nevertheless, it was not able to compensate for the reduction in sperm viability and motility following cryopreservation. Opposite results were obtained with Compound C. Therefore, the authors hypothesized that RSV-induced improvement of cryopreserved sperm functions could be mediated through activation of AMPK. The same authors investigated the effect of 15 µM of RSV on sperm DNA integrity and fertilizing capacity by quantifying the presence of key paternal transcripts considered as potential markers for male fertility (protamine 1 (PRM1) and protamine 2(PRM2)) and pregnancy success (adducin 1 alpha (ADD1)) in cryopreserved human spermatozoa [49]. They showed that the addition of RSV to the freezing medium significantly decreased sperm DNA fragmentation compared with the control group. Furthermore, a statistically significant amelioration in terms of mRNA expression of PRM, PRM2, and ADD1 was found compared to controls. RSV-induced AMPK activation could be responsible for these effects since the use of Compound C (30 µM) induced opposite results. The authors hypothesized that stimulation of AMPK activity by RSV could stabilize the transcripts tested in this study by improving the interaction among mRNAs, making mRNA molecules more resistant to cryopreservation.

In 2020, the Iranian group of Mohammadzadeh and his collaborators published a study that tested the post-thawing effect of RSV directly on spermatozoa separated using the swim-up method [50]. Specifically, the samples were obtained by 10 normozoospermic men and 10 asthenozoospermic patients. After swim-up, spermatozoa were divided into two aliquots: the experimental one that was added with 30 μmol/L of RSV, and the control sample incubated without RSV. The authors found that, despite the protective effects on the semen samples after freezing, RSV did not significantly influence sperm parameters and chromatin quality in normozoospermic men and asthenozoospermic patients. However, they found a better sperm chromatin quality in the RSV treated group than in the control one, although this effect was more evident before freezing rather than after, likely due to the increased ROS production in the last condition. Therefore, the authors concluded that the results obtained in previous studies, conducted on whole semen samples, were in part affected by strong antioxidant compounds naturally contained in seminal fluid, which could better neutralize the effects of oxidative stress [50].

Table 1 summarizes the principal studies evaluating the effects of RSV on sperm cryopreservation.

## 5. Resveratrol and Fertility in Obese Male Patients

Animal models [54,55,56] suggest that RSV could have a protective role in sperm function in obese men. Overweight and obesity, in fact, have been shown to increase the risk of developing reproductive disorders [54,57]. Many studies studied have shown that sperm quality is significantly affected by overweight, in terms of decreased sperm concentration and total motile sperm count [54,55,56].

Many preclinical studies have reported that RSV improves glucose metabolism and homeostasis, thus, suggesting its possible use as a therapeutic strategy in the fight against obesity [35]. However, data from human studies are still scarce and, at times, contradictory and inconclusive. Based on these data, it can be hypothesized that RSV may be effective for the treatment of infertility in obese males.

Cui and collaborators [51] incubated 60 semen samples obtained from obese and asthenozoospermic patients with 0–100 µM RSV for 30 min and observed various degrees of improvement in sperm motility. Among the various concentrations of RSV tested in this study, 30 µM/L was associated with the most significant improvement in sperm motility when compared with controls. The authors also evaluated the effect of this concentration of RSV on seminal plasma zinc concentration and spermatozoa acrosin activity. The RSV-treated group showed a significantly increased seminal plasma zinc concentration and sperm acrosin activity compared with the control group. They concluded that RSV could have therapeutic and protective effects against obesity-induced semen and sperm parameter abnormalities (Table 1) [51]. To the best of our knowledge, this is the only study conducted on human spermatozoa so far, while many studies have been conducted on animal models [58,59]. Therefore, further studies will be needed to evaluate the effective efficacy of RSV in this selected category of patients.

## 6. Summary and Conclusions

RSV is a natural non-flavonoid polyphenol compound derived from plants, which shows protective effects in humans. Many studies, based on animal models, have shown a positive role of RSV in male fertility, giving a wide motivation to explore RSV’s role in human beings. Studies conducted on human spermatozoa have shown that RSV at low concentrations has a positive effect on sperm motility, whereas at higher concentrations it has a detrimental effect on this parameter [34,35,36,37,38]. Thanks to its scavenger activity, RSV has a protective role against sperm DNA damage caused by oxidative stress, so it could have a therapeutic effect in men with idiopathic infertility [38,39,40,41]. A pilot trial has also shown that nutraceuticals containing RSV have a positive effect on sperm concentration, total sperm count, and both total and progressive motility, without affecting sperm morphology, ejaculate volume, and pH [14]. The anti-oxidative properties of RSV make it useful in preventing sperm damage induced by freezing during cryopreservation. Studies have shown that RSV added to semen samples after freezing was able to significantly preserve DNA and lipid damage in both infertile patients and fertile men, but it was not able to prevent the post-freezing motility decrease [45,46,47]. The mechanism by which RSV improves cryopreserved sperm functions may relate to activation of AMPK, which is mainly localized in the post-equatorial region of the sperm head and the entire flagellum [34,35,36,37,38,39,40,41,42,43,44,45,46,47,48,49,52,53]. Despite the protective effects of RSV on semen samples, RSV has not been able to significantly affect sperm parameters and chromatin quality in spermatozoa separated by swim-up which do not have an additive effect of antioxidants from the semen. Moreover, RSV seems to have a therapeutic effect against fertility disorders related to obesity.

The lack of consistent data strongly suggests that there is a need for further study on the human models necessary to assess the efficacy of RSV in the treatment of male infertility and specifically in the subpopulation of patients with obesity.

## Figures and Tables

**Table 1 molecules-26-02495-t001:** Summary of the main studies on the effects of resveratrol on male fertility.

Author and Year of Publication	Type of Study	Patients	Resveratrol Dose(s) Used	Duration of Resveratrol Administration	Main Findings
Collodel et al., 2011 [38]	In vitro	Non-reported	6, 15, 30, 50, 100 µM	60 min	Highest sperm motility obtained with 6 and 15 µM Motility completely lost with the concentration of 100 µM Cytotoxic effect at higher doses Protective effect against LPO induced by oxidative stress with 15 µM but not with 6 µM Effect in preserving sperm chromatin texture but not acrosome Possible role as a ROS scavenger
Nashtaei et al., 2016 [34]	In vitro	22 donors with normozoospermia and proven fertility	5, 15, 25 µM	30 or 60 min	Reduction in total and progressive sperm motility only with higher concentration (25 µM) AMPK expressed in human fresh spermatozoa (in the post-equatorial region of the sperm head and in the entire flagellum) Increased AMPK phosphorylation and MMP, while decreased ROS and apoptosis-like changes in frozen–thawed spermatozoa after incubation with RSV Motility and viability not ameliorated after incubation with RSV RSV-induced improvement of cryopreserved sperm functions mediated through activation of AMPK
Alamo et al., 2019 [41]	In vitro	30 healthy men with normal sperm parameters	15 µM/mL	30 min	Decreased percentage of spermatozoa with abnormal chromatin compactness No significant effect on LPO Decreased percentage of spermatozoa with elevated mitochondrial superoxide production Reverted the detrimental effects of BaP on chromatin compactness and LPO Possible therapeutic role against idiopathic infertility
Illiano et al., 2020 [14]	Prospective single-center clinical study (in vivo study)	20 patients with oligozoospermia and/or asthenozoospermia	150 mg every 12 h per os	Follow-up at 1, 3 and 6 months after treatment	Improvement of sperm concentration, total sperm count, total and progressive motility No changes in sperm morphology, pH, and volume
Branco et al., 2010 [45]	In vitro	10 infertile patients and 10 healthy donors of proven fertility	0.1, 1, 10 mM	Non-reported	Decrease of DNA damage in both infertile patients and fertile men with 10 mM, but not with 0.1 or 1 mM No effect against the post-freezing motility decrease
Garcez et al., 2010 [46]	In vitro prospective study	20 infertile patients and 10 healthy donors of proven fertility	0.1, 1, 10 mM	60 min	Decrease of sperm lipid damage induced by cryopreservation in both fertile men and infertile patients No effect against the reduction of sperm motility Effect dose-independent
Meamar et al., 2012 [47]	In vitro prospective study	21 donors with normozoospermia	10, 100, 1000 µM	3 days	Dose-dependent effects of RSV addition to cryopreservation medium Decrease of post-thawing sperm SDF levels with 10 and 100 µM without effect on motility or viability Reduction of motility and viability at the RSV concentration of 1 mM Cytotoxic effect at higher concentration also after cryopreservation
Li and al., 2018 [48]	In vitro	50 donors with normozoospermia and 50 patients with OAT	30 μmol/L	Non-reported	Increased sperm progressive motility, total motility, and viability after cryopreservation
Nashtaei et al., 2018 [49]	In vitro	22 healthy volunteers with proven fertility	15 µM	Non-reported	Decreased DNA fragmentation after addition of RSV to the freezing medium Improvement of mRNA expression levels of PRM, PRM2, and ADD1 (markers for male fertility and pregnancy success)
Mohammadzadeh et al., 2020 [50]	In vitro	10 donors with normozoospermia and 10 men with asthenozoospermia	30 µmol/L	60 min	No significant effect on sperm parameters and chromatin quality
Cui et al., 2016 [51]	In vitro prospective case-control study	60 obese patients	2.6, 6, 15, 30, 50, 100 µmol/L	30 min	Treatment with 30 µM/L obtained the most significant improvement in sperm motility Increased of both seminal plasma zinc concentration and sperm acrosin activity RSV could have therapeutic and protective effects against obesity-induced abnormal sperm parameters

**Abbreviations (in alphabetical order):** ADD = adducing; AMPK = adenosine monophosphate-activated protein kinase; BaP = benzo-α-pyrene; LPO = lipid peroxidation; MMP = mitochondrial membrane potential; OAT = oligoasthenoteratozoospermia; PRM = protamine; ROS = reactive oxygen species; RSV = resveratrol; SDF = sperm DNA fragmentation.

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
