# Peer review of "The Role of Resveratrol in Human Male Fertility"

_molecules, 2021, doi:10.3390/molecules26092495_

Round 1

Reviewer 1 Report

Please check the authors' instruction. You had some mistakes in text headings.

This review contains a large number of references related to previous literature and more new references are needed.

What order does the literature appear to be in Table 1, it looks confusing?

L41-42: References are needed here.

L98-100: References are needed here.

L103-112: Error in citation of reference.

L112-113: Here are some studies on the mouse, with suggested extended references.

  1. de Oliveira FA, Costa WS, Sampaio FJB and Gregorio BM, Resveratrol attenuates metabolic, sperm, and testicular changes in adult Wistar rats fed a diet rich in lipids and simple carbohydrates. Asian Journal of Andrology 21:201-207 (2019).
  2. Lv ZM, Ling MY and Chen C, Comparative proteomics reveals protective effect of resveratrol on a high-fat diet-induced damage to mice testis. Syst Biol Reprod Med 66:37-49 (2020).

Author Response

This review contains a large number of references related to previous literature and more new references are needed.

Comment 1: What order does the literature appear to be in Table 1, it looks confusing?

Answer to comment 1: In the original version of the manuscript, Table 1 was reported at the end of the review. The references appear following the order in which the studies were cited in the text. For this reason, if Editor agrees, we suggest to place the table in its originally proposed position.

Comment 2: L41-42: References are needed here.

Answer to comment 2: We added a suitable reference.

Comment 3: L98-100: References are needed here.

Answer to comment 3: We added a suitable reference.

Comment 4: L103-112: Error in citation of reference.

Answer to comment 4: The mistake was corrected.

Comment 5: L112-113: Here are some studies on the mouse, with suggested extended references.

  • de Oliveira FA, Costa WS, Sampaio FJB and Gregorio BM, Resveratrol attenuates metabolic, sperm, and testicular changes in adult Wistar rats fed a diet rich in lipids and simple carbohydrates. Asian Journal of Andrology 21:201-207 (2019).
  • Lv ZM, Ling MY and Chen C, Comparative proteomics reveals protective effect of resveratrol on a high-fat diet-induced damage to mice testis. Syst Biol Reprod Med 66:37-49 (2020).

Answer to comment 5: The references suggested were discussed and included in the revised version of the manuscript.

All the changes made in the text are highlighted in pink for an easier identification.

Reviewer 2 Report

 In this manuscript, the authors summarize the effects of resveratrol (RSV) on human male fertility and the potential mechanisms underlying its effects. RSV is a type of non-flavonoid polyphenol that is naturally produced by numerous plants, such as grapes, pistachios and peanuts. The main criticism of the current manuscript is its inaccurate information regarding reference publications studying RSV effects on human spermatozoa. As the submitted review article is aimed at evaluating the effects of RSV on male fertility based on the outcomes reported in previously conducted studies, the inexact description about the study designs and their results could mislead readers.

Major comments:

1.  Throughout the manuscript and Table 1, the introduced publications are not correctly described. Representative examples are listed below. The authors have to re-evaluate all original publications presented in the current manuscript to correct erroneous points.

  • The study performed by Collodel et al., (2011) is introduced on page 4, line 115 (Reference #38) and in Table 1. Although the original paper shows that the RSV treatment was performed at different concentrations for 1 hour at 37C, Table 1 describes that the duration of RSV administration is not reported.
  • Similarly, the study introduced as Reference #52 on page 14, line 77 and in Table1 reports that RSV administration for spermatozoa was performed at a dose of 30umol/l for 1hr of incubation at 37C. However, Table 1 describes that the treatment duration is not reported.
  • Reference #41 is introduced on page 5, line 138, where the authors of this study reported RSV treatment at the concentration of 15 uM/mLwas performed. However, the current manuscript describes the dose used in this study is 15 M/mL (page 6, line 155).
  • The study conducted by Illiano et al (Reference #14) is presented on page 6, line 163 and in Table 1. Although the original paper reported that patients received a daily consumption of 150 mg of RSV by taking a total of 2 tablets per day, the current manuscript describes that a dose of 300 mg of RSV was administered to patients, which is double the amount of the dose used.
  • On page 15, line 103, the reference study is introduced as Reference#57, describing that Clui and collaborators performed an RSV treatment study using semen samples obtained from obese and asthenozoospermic patients. According to the context, the reference number and the name of the first author in the text body are incorrect. The reference number should be #58, and the first author should be Cui.

2.  It is difficult to link the information indicated in Table 1 and the descriptions in the manuscript. The authors need to show reference numbers in Table1. It is also necessary to match the year of publication shown in Table 1 and the publication information listed in the Reference.

3.  It is highly recommended that the current table be separated into two different tables based on the two different contents: the studies about the effects of RSV on human sperm parameters, and the studies about the effects of RSV on sperm cryopreservation.

Minor comments:

  • The note under the Table 1 should be Abbreviation, not Legend.
  • The description indicated on page 13, line 41 “10 vs. 10 mM” does not make sense; it should be “10 uM vs. 10 mM”.
  • Please delete the repeated ‘Firstname Lastname’ from the author list.

Author Response

In this manuscript, the authors summarize the effects of resveratrol (RSV) on human male fertility and the potential mechanisms underlying its effects. RSV is a type of non-flavonoid polyphenol that is naturally produced by numerous plants, such as grapes, pistachios and peanuts. The main criticism of the current manuscript is its inaccurate information regarding reference publications studying RSV effects on human spermatozoa. As the submitted review article is aimed at evaluating the effects of RSV on male fertility based on the outcomes reported in previously conducted studies, the inexact description about the study designs and their results could mislead readers.

Major comments:

Comment 1: Throughout the manuscript and Table 1, the introduced publications are not correctly described. Representative examples are listed below. The authors have to re-evaluate all original publications presented in the current manuscript to correct erroneous points.

The study performed by Collodel et al., (2011) is introduced on page 4, line 115 (Reference #38) and in Table 1. Although the original paper shows that the RSV treatment was performed at different concentrations for 1 hour at 37C, Table 1 describes that the duration of RSV administration is not reported.

Similarly, the study introduced as Reference #52 on page 14, line 77 and in Table1 reports that RSV administration for spermatozoa was performed at a dose of 30 umol/l for 1hr of incubation at 37C. However, Table 1 describes that the treatment duration is not reported.

Reference #41 is introduced on page 5, line 138, where the authors of this study reported RSV treatment at the concentration of 15 uM/mL was performed. However, the current manuscript describes the dose used in this study is 15 M/mL (page 6, line 155).

The study conducted by Illiano et al (Reference #14) is presented on page 6, line 163 and in Table 1. Although the original paper reported that patients received a daily consumption of 150 mg of RSV by taking a total of 2 tablets per day, the current manuscript describes that a dose of 300 mg of RSV was administered to patients, which is double the amount of the dose used.

On page 15, line 103, the reference study is introduced as Reference#57, describing that Clui and collaborators performed an RSV treatment study using semen samples obtained from obese and asthenozoospermic patients. According to the context, the reference number and the name of the first author in the text body are incorrect. The reference number should be #58, and the first author should be Cui.

Answer to comment 1: Thank you for these comments and the time you spent on this. All the references indicated were carefully reviewed and the mistakes corrected. Illiano and colleagues in their study administered 150 mg of resveratrol every 12 hours, for a total dosage of 300 mg/day. We clarified it in the table.

 Comment 2: It is difficult to link the information indicated in Table 1 and the descriptions in the manuscript. The authors need to show reference numbers in Table1. It is also necessary to match the year of publication shown in Table 1 and the publication information listed in the Reference.

Answer to comment 2: We added references in the Table and the year of publication was corrected.

Comment 3: It is highly recommended that the current table be separated into two different tables based on the two different contents: the studies about the effects of RSV on human sperm parameters, and the studies about the effects of RSV on sperm cryopreservation.

Answer to comment 3: We agree with your comment. In the original version of the manuscript, Table 1 was reported at the end of the review. Some studies are cited in different paragraphs. For this reason, we elaborated on a single table. We would like to live a single table to be placed at the end of the review because we think this will be clearer for the reader. However, if the Editor feels differently, we can split the table but in this case, some studies will be reported twice.

Minor comments:

Comment 4: The note under the Table 1 should be Abbreviation, not Legend.

Comment 5: The description indicated on page 13, line 41 “10 vs. 10 mM” does not make sense; it should be “10 uM vs. 10 mM”.

Comment 6: Please delete the repeated ‘Firstname Lastname’ from the author list.

Answer to comments 4, 5, and 6: All these mistakes were corrected.

Round 2

Reviewer 2 Report

In the revised manuscript, the authors’ points are clear and supported by appropriate reference publications and their descriptions, which will clarify readers’ understanding. I recommend the manuscript for publication after the following minor revisions.

The current Table1 could lead to confusion for readers. Please specifically state the study style for each publication.

  • Except for the publication by Illiano et al. (Reference #14), the studies introduced in Table 1 were performed with in-vitro experimental designs, where the collected sperm from patients were treated by a certain concentration of resveratrol. Whereas, in the study by Illiano et al., the patients with oligozoosperima and/or asthenozoospermia took an oral tablet containing resveratrol, and the effect of treatment was evaluated by a semen analysis after the intervention. Please add a simple explanation regarding the study design of Reference #14 in Table 1 to clarify this point.
  • The publication by Cui et al. (Reference #59) is described as ‘Prospective case control study’. To have a consistent format, the authors need to describe this study type as ‘In-vitro prospective case control study’ in Table 1.

Author Response

In the revised manuscript, the authors’ points are clear and supported by appropriate reference publications and their descriptions, which will clarify readers’ understanding. I recommend the manuscript for publication after the following minor revisions.

The current Table1 could lead to confusion for readers. Please specifically state the study style for each publication.

  • Except for the publication by Illiano et al. (Reference #14), the studies introduced in Table 1 were performed with in-vitro experimental designs, where the collected sperm from patients were treated by a certain concentration of resveratrol. Whereas, in the study by Illiano et al., the patients with oligozoosperima and/or asthenozoospermia took an oral tablet containing resveratrol, and the effect of treatment was evaluated by a semen analysis after the intervention. Please add a simple explanation regarding the study design of Reference #14 in Table 1 to clarify this point.
  • The publication by Cui et al. (Reference #59) is described as ‘Prospective case control study’. To have a consistent format, the authors need to describe this study type as ‘In-vitro prospective case control study’ in Table 1.

Answer:  Thank you for your comments. We modified Table 1 as you suggested. 

The last changes made in the table are highlighted in yellow for an easier identification.